# HyNet: Learning Local Descriptor with Hybrid Similarity Measure and Triplet Loss

Yurun Tian[1]    Axel Barroso-Laguna[1]    Tony Ng[1]    Vassileios Balntas[2]
Krystian Mikolajczyk[1]

[1] Imperial College London
[2] Facebook Reality Labs
{yurun.tian,axel.barroso17,tony.ng14,k.mikolajczyk}@imperial.ac.uk
vassileios@fb.com

## Abstract

Recent works show that local descriptor learning benefits from the use of $L_2$ normalisation, however, an in-depth analysis of this effect lacks in the literature. In this paper, we investigate how $L_2$ normalisation affects the back-propagated descriptor gradients during training. Based on our observations, we propose HyNet, a new local descriptor that leads to state-of-the-art results in matching. HyNet introduces a hybrid similarity measure for triplet margin loss, a regularisation term constraining the descriptor norm, and a new network architecture that performs $L_2$ normalisation of all intermediate feature maps and the output descriptors. HyNet surpasses previous methods by a significant margin on standard benchmarks that include patch matching, verification, and retrieval, as well as outperforming full end-to-end methods on 3D reconstruction tasks. Codes and models are available at https://github.com/yuruntian/HyNet.

## 1   Introduction

Local feature detectors and descriptors play a key role in many computer vision tasks such as 3D reconstruction[39], visual localisation[38, 42] and image retrieval[1, 33, 29]. Recently, joint detection and description [51, 30, 10, 11, 34, 22, 25, 13, 45, 5] has drawn significant attention. Despite the alluring idea of the end-to-end detection and description, the classic two-stage strategy withstood years of tests in many computer vision tasks and still gives a competitive performance in standard benchmarks [7, 2, 40, 18]. Moreover, customised matchers [27, 36, 35, 6, 37] have also contributed to boosting the matching performance, where the time complexity is critical. Despite the progress in end-to-end methods, the two-stage process still deserves attention since it often leads to competitive results in the overall matching system.

Deep descriptors [43, 3, 46, 26, 19, 15, 47, 54, 53] have shown superiority over hand-crafted ones [23, 50] in different tasks [2, 18, 7, 40]. Current works mainly focus on improving the loss function or the sampling strategy. L2-Net [46] introduces a progressive batch sampling with an N-Pair loss. HardNet [26] uses a simple yet effective hard negative mining strategy, justifying the importance of the sampling. Other than contrastive or triplet loss, DOAP [15] employs a retrieval based ranking loss. GeoDesc [24] integrates geometry constraints from multi-view reconstructions to benefit the training. Besides the first-order optimisation, SOSNet [47] shows that second-order constraints further improve the descriptors.

It has been widely observed that $L_2$ normalisation of the descriptors leads to consistent improvements. Methods such as [46, 26, 15, 12, 47, 56, 54] which $L_2$ normalised descriptors, significantly outperform early unnormalised descriptors [43, 3]. Moreover, even hand-crafted descriptors can be improved with $L_2$ normalisation [2]. All such observations indicate that descriptors are better distinguished by their vector directions rather than the magnitudes ($L_2$ norms), where similar conclusions can also be found in other feature embedding tasks [49, 9, 21].

We therefore analyse the impact of $L_2$ normalisation on learning from the gradients perspective. Since the gradients for each layer are generated via the chain rule [14], we analyse them at the beginning of the chain, where they are generated by the given similarity measure or distance metric. Our intuition is that the gradient direction should benefit the optimisation of descriptor directions, while the gradient magnitude should be adaptive to the level of hardness of the training samples. Consequently, HyNet is introduced to make better use of the gradient signals in terms of direction and magnitude.

Despite the evolving design of loss function, triplet loss is still employed in state-of-the-art local descriptors [26, 47]. Furthermore, triplet loss has also earned noticeable popularity in various embedding tasks, *e.g*, face recognition [41, 31] and person re-identification [8, 16]. An interesting observation in [28] indicates that the improvements from the classic contrastive and triplet loss are marginal. In this work, we further show that state-of-the-art local descriptor can be learned by triplet loss with a better designed similarity measure.

Specifically, we propose: 1) a hybrid similarity measure that can balance the gradient contributions from positive and negative samples, 2) a regularisation term which provides suitable constraints on descriptor norms, and 3) a new network architecture that is able to $L_2$ normalise the intermediate feature maps.

## 2 Gradient Analysis

In this section, we explore how the widely used inner product and $L_2$ distance provide gradients for training normalised and unnormalised descriptors.

### 2.1 Preliminaries

We denote $\mathcal{L}(\psi(\mathbf{x}, \mathbf{y}))$ as the loss for a descriptor pair $(\mathbf{x}, \mathbf{y})$, where $\psi(\cdot, \cdot)$ can be a similarity measure or a distance metric. To ensure consistency in the following of the paper, we refer to distance metric also as a similarity measure even though it measures the inverse similarity. Whether $(\mathbf{x}, \mathbf{y})$ are positive (matching) or negative (non-matching), the gradients with respect to the descriptors are calculated as:

$$\frac{\partial \mathcal{L}}{\partial \mathbf{x}} = \frac{\partial \mathcal{L}}{\partial \psi} \frac{\partial \psi}{\partial \mathbf{x}}, \quad \frac{\partial \mathcal{L}}{\partial \mathbf{y}} = \frac{\partial \mathcal{L}}{\partial \psi} \frac{\partial \psi}{\partial \mathbf{y}}, \tag{1}$$

where $(\mathbf{x}, \mathbf{y})$ are omitted for clarity. Importantly, the gradients for learnable weights of a network are derived in Eqn.(1) at the beginning of the chain, and play a key role during training. Note that $\frac{\partial \mathcal{L}}{\partial \psi}$ is a scalar, while the direction of the gradient is determined by the partial derivatives of $\psi$. We consider the most commonly used inner product and $L_2$ distance, for descriptors with and without $L_2$ normalisation:

$$\bar{s} = \mathbf{x}^{\mathrm{T}}\mathbf{y}, \quad s = \frac{\mathbf{x}^{\mathrm{T}}\mathbf{y}}{\|\mathbf{x}\|\|\mathbf{y}\|}, \quad \bar{d} = \|\mathbf{x} - \mathbf{y}\|, \quad d = \|\frac{\mathbf{x}}{\|\mathbf{x}\|} - \frac{\mathbf{y}}{\|\mathbf{y}\|}\|, \tag{2}$$

where $\| \cdot \|$ denotes the $L_2$ norm ($\|\mathbf{x}\| = \sqrt{\sum \mathbf{x}_i^2}$). $\bar{s}$ and $\bar{d}$ are inner product and $L_2$ distance for raw descriptors while $s$ and $d$ are for normalised ones. Note that we consider $L_2$ normalisation as a part of the similarity measure.

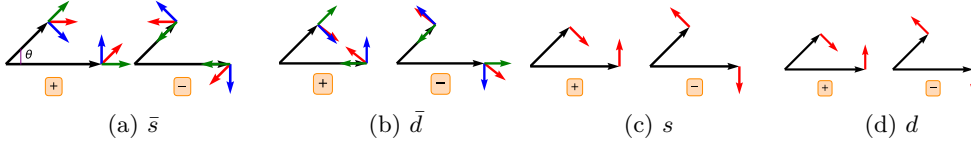

(a) $\bar{s}$        (b) $\bar{d}$        (c) $s$        (d) $d$

Figure 1: Gradient descent directions derived in Eqn. 3, with $+$ and $-$ for positive and negative pairs. $\theta$ is the angle between the descriptors. **Black** arrows: descriptors before $L_2$ normalisation. **Red** arrow: gradient descent direction from $\boldsymbol{\Delta}$. **Green** arrow: parallel component from $\boldsymbol{\Delta}_{\parallel}$. Blue arrows: orthogonal component from $\boldsymbol{\Delta}_{\perp}$. Better viewed in colour.

We then obtain the partial derivatives:

$$\frac{\partial \bar{s}}{\partial \mathbf{x}} = \mathbf{y}, \quad \frac{\partial \bar{s}}{\partial \mathbf{y}} = \mathbf{x}, \quad \frac{\partial \bar{d}}{\partial \mathbf{x}} = \frac{1}{\bar{d}}(\mathbf{x} - \mathbf{y}), \quad \frac{\partial \bar{d}}{\partial \mathbf{y}} = \frac{1}{\bar{d}}(\mathbf{y} - \mathbf{x}),$$

$$\frac{\partial s}{\partial \mathbf{x}} = \frac{1}{\|\mathbf{x}\|\|\mathbf{y}\|}(\mathbf{y} - \frac{\mathbf{x}^{\mathrm{T}}\mathbf{y}}{\|\mathbf{x}\|^2}\mathbf{x}), \quad \frac{\partial s}{\partial \mathbf{y}} = \frac{1}{\|\mathbf{x}\|\|\mathbf{y}\|}(\mathbf{x} - \frac{\mathbf{x}^{\mathrm{T}}\mathbf{y}}{\|\mathbf{y}\|^2}\mathbf{y}), \quad (3)$$

$$\frac{\partial d}{\partial \mathbf{x}} = \frac{1}{d\|\mathbf{x}\|\|\mathbf{y}\|}(\frac{\mathbf{x}^{\mathrm{T}}\mathbf{y}}{\|\mathbf{x}\|^2}\mathbf{x} - \mathbf{y}), \quad \frac{\partial d}{\partial \mathbf{y}} = \frac{1}{d\|\mathbf{x}\|\|\mathbf{y}\|}(\frac{\mathbf{x}^{\mathrm{T}}\mathbf{y}}{\|\mathbf{y}\|^2}\mathbf{y} - \mathbf{x}).$$

In the following sections we analyse the above gradients in terms of directions and magnitudes.

## 2.2 Gradient Direction

Optimal gradient direction is the key for convergence, *i.e.*, a learning process will not converge given incorrectly directed gradients, regardless of the learning rate. We denote $\boldsymbol{\Delta} = \boldsymbol{\Delta}_{\parallel} + \boldsymbol{\Delta}_{\perp}$, where $\boldsymbol{\Delta}$ is the overall gradient direction, $\boldsymbol{\Delta}_{\parallel}$ and $\boldsymbol{\Delta}_{\perp}$ are the parallel and orthogonal components respectively. According to Eqn. (3), we obtain $|\boldsymbol{\Delta}_{\parallel}| = \mathbf{x}^{\mathrm{T}}\frac{\partial \bar{s}}{\partial \mathbf{x}} = 0$, and similarly for $\mathbf{y}^{\mathrm{T}}\frac{\partial \bar{s}}{\partial \mathbf{y}} = 0$, $\mathbf{x}^{\mathrm{T}}\frac{\partial \bar{d}}{\partial \mathbf{x}} = 0$, and $\mathbf{y}^{\mathrm{T}}\frac{\partial \bar{d}}{\partial \mathbf{y}} = 0$, *i.e.*, gradients are always orthogonal to the descriptors, indicating that $L_2$ normalised descriptors only have $\boldsymbol{\Delta}_{\perp}$. Meanwhile, both components of unnormalised descriptors are non-zero. For better understanding, we illustrate 2D descriptors and the corresponding gradient descent directions (negative gradient direction) in Fig. 1, where $\theta$ is the angle between descriptors. Specifically, $\boldsymbol{\Delta}_{\parallel}$ modifies the descriptor magnitude ($L_2$ norms), while $\boldsymbol{\Delta}_{\perp}$ updates the descriptor direction. However, since descriptor magnitudes can be harmful for matching (see Sec. 1), the training should focus on the optimisation of the descriptor directions, which can be achieved with $L_2$ normalised descriptors. An interesting question is whether it is possible to make a better use of $\boldsymbol{\Delta}_{\parallel}$. We address this problem in Sec. 3.1 and show that detailed analysis leads to a training constraint that improve the performance.

## 2.3 Gradient Magnitude

The training gradients should have not only the optimal directions but also the properly scaled magnitudes. The magnitude should be adapted to the level of 'hardness' of the training samples, *i.e.*, hard samples should receive a stronger update over easy ones.

We focus on $L_2$ normalised descriptors whose gradients have optimal directions. We denote $\mathbf{u} = \frac{\mathbf{x}}{\|\mathbf{x}\|}$ and $\mathbf{v} = \frac{\mathbf{y}}{\|\mathbf{y}\|}$ as two descriptors normalised with $L_2$. Further, $s$ and $d$ are expressed as a function of the angle between the descriptors:

$$s(\theta) = \mathbf{u}^{\mathrm{T}}\mathbf{v} = \cos\theta, \quad g_s(\theta) = |s'(\theta)| = |\sin\theta|,$$

$$d(\theta) = \|\mathbf{u} - \mathbf{v}\| = \sqrt{2(1 - \cos\theta)}, \quad g_d(\theta) = |d'(\theta)| = |\frac{\sin\theta}{\sqrt{2(1 - \cos\theta)}}|, \quad (4)$$

where $\theta = \arccos \mathbf{u}^{\mathrm{T}}\mathbf{v}$, $g(\theta)$ is the gradinet magnitude and $|\cdot|$ denotes absolute value operator.

We analyse the gradient magnitudes from Eqn. (4) in the real descriptor space during training. Fig. 2(a) shows the distribution of $\theta$ from 512K descriptor pairs, where the number of positive and negative pairs is 50% each. Following the hard negative mining strategy of [26], we

sample 512 triplets (one positive pair and one negative) from each of the 1K randomly constructed batches of size 1024. Fig. 2(a) shows the $\theta$ distribution of HardNet and SOSNet in training, *i.e.*, both models are trained and tested on *Liberty*. Note that from Eqn. 4 the gradient magnitudes are periodic functions with a period of $\pi$. As shown, almost all hard negatives and positives have $\theta$ in the range $[0, \pi/2]$. Therefore, we observe how $g_s$ and $g_d$ behave in range $[0, \pi/2]$, which is highlighted in Fig. 2(b).

The gradients differ, *i.e.*, $g_s$ is monotonically increasing while $g_d$ is decreasing. It indicates that $g_s$ is more beneficial for the optimisation of positives, since hard positives (large $\theta \to \pi/2$), generate large gradients compared to easy positives (small $\theta$). In contrast, $g_d$ favours negatives, as hard negatives (small $\theta$) generate large updates compared to the easy negatives (large $\theta$). These observations lead to the conclusion that neither the inner product nor the $L_2$ on its own can balance the optimisation with positives and negatives.

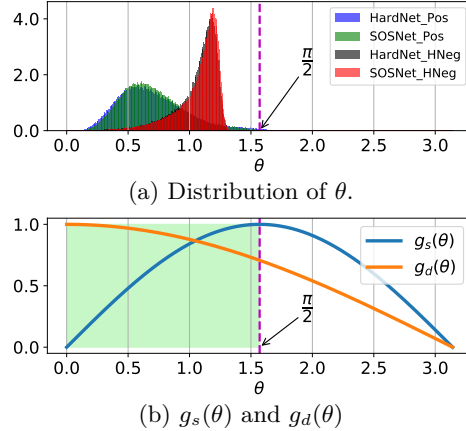

(a) Distribution of $\theta$.

(b) $g_s(\theta)$ and $g_d(\theta)$

Figure 2: Gradient magnitude and distribution of $\theta$.

It is also worth noting that according to Eqn. (1), the overall gradient magnitude is further weighted by $\frac{\partial \mathcal{L}}{\partial s}$, which means a better form of $\mathcal{L}$ may alleviate the inherent flaws of $g_s$ and $g_d$. Consequently, in Sec. 3.2 we show that a carefully designed similarity measure leads to the state-of-the-art performance with the standard triplet loss.

## 3   Method

Building upon the analysis from the previous section, we propose to improve the descriptor learning by 1) introducing a regularisation term that provides a beneficial $\boldsymbol{\Delta}_{\parallel}$, 2) a hybrid similarity measure that can strike a balance between the contribution of positives and negatives to the gradient update, 3) a new network architecture that normalises the intermediate feature maps mimicking the output descriptors such that they are optimised in their directions rather than the magnitudes.

### 3.1   $L_2$ Norm Regularisation

Sec. 2.2 shows that $L_2$ normalisation excludes parallel gradients $\boldsymbol{\Delta}_{\parallel}$, *i.e.*, there are no constraints on the descriptor norms which can vary with scaling of image intensities. Intuitively, a possible way of making positive contributions from $\boldsymbol{\Delta}_{\parallel}$ to the optimisation is to introduce the following constraint before the $L_2$ normalisation:

$$R_{L_2} = \frac{1}{N} \sum_{i=1}^{N} (\|\mathbf{x}_i\| - \|\mathbf{x}_i^+\|)^2, \tag{5}$$

where $\mathbf{x}_i$ and $\mathbf{x}_i^+$ are a positive pair of descriptors before $L_2$ normalisation. As a regularisation term, $R_{L_2}$ drives the network to be robust to image intensity changes, *e.g*, caused by different illuminations.

### 3.2   Hybrid Similarity Measure and Triplet Loss

Recent efforts on improving the standard triplet loss include smart sampling of triplets [26, 52] and adaptive margin [55, 55]. In contrast, we explore to boost the triplet loss with a hybrid similarity measure such that better gradients can be generated. As discussed in Sec. 2.3, $s$ and $d$ favours the positive and negative samples respectively, therefore we propose a hybrid

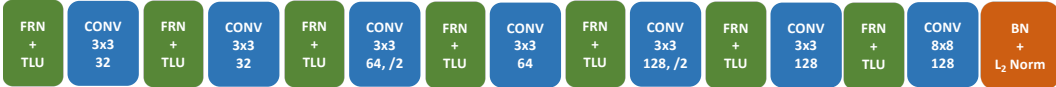

Figure 3: HyNet architecture.

similarity measure $s_H$ that can make a balance between them:

$$\mathcal{L}_{Triplet} = \frac{1}{N}\sum_{i=1}^{N}\max(0, m + s_H(\theta_i^+) - s_H(\theta_i^-)),$$

$$s_H(\theta) = \frac{1}{Z}[\alpha(1 - s(\theta)) + d(\theta)], \tag{6}$$

where $\alpha$ is a scalar ranging from 0 to $+\infty$ adjusting the ratio between $s$ and $d$, and $Z$ is the normalising factor ensuring the gradient has the maximum magnitude of 1.

From the gradient perspective, when the margin constraint in Eqn. (6) is not satisfied, we obtain $\frac{\partial\mathcal{L}_{Triplet}}{\partial s_H(\theta_i^+)} = \frac{\partial\mathcal{L}_{Triplet}}{\partial s_H(\theta_i^-)} = 1$, otherwise 0. Hence, $s_H'(\theta_i^+)$ and $s_H'(\theta_i^-)$ are directly related to the gradient magnitude. We will show in Sec. 5 that Eqn. (6) performs better over other possible solutions for balancing the gradients. Finally, our overall loss function is defined as:

$$\mathcal{L} = \mathcal{L}_{Triplet} + \gamma R_{L_2}, \tag{7}$$

where $\gamma$ as a regularisation parameter and $\alpha$ balancing the contributions from $s$ and $d$. Optimal $\alpha$ can be found by a grid search which is discussed in Sec. 5.

### 3.3 Network Architecture

In the work of L2-Net [46], the authors show that flattened feature maps can be optimised in the same way as the final descriptors. Thus, we are inspired to generalise the observations of Sec.2 to the intermediate feature maps. Instead of building extra loss functions, we propose to better manipulate the gradients for different layers. Since feature maps are also feature vectors in high dimensional spaces, the previous gradient analysis can still be applied. Our goal is to generate orthogonal gradients for the feature maps of all layers by $L_2$ normalising them, such that they can be better optimised in terms of directions mimicking the descriptors. To this end, we can directly adopt the off-the-shelf Filter Response Normalisation(FRN) [44], which has been recently proposed and shown promising results in the classification task. The core idea of FRN is to $L_2$ normalise the intermediate feature maps with learnable affine parameters. Specifically, FRN normalises each layer of feature maps by:

$$\hat{\mathbf{f}}_i = \gamma\sqrt{N}\frac{\mathbf{f}_i}{\|\mathbf{f}_i\|} + \beta, \tag{8}$$

where $\gamma$ and $\beta$ are learned parameters, $\mathbf{f}_i$ is the flattened feature map of the $i$-th channel and $N$ is the number of pixels. Note that, it is also argued in [44] that after FRN the gradients w.r.t. $\mathbf{f}_i$ are always orthogonal, which suits our scenario. We will show in Sec. 5 that although FRN can provide general performance boost, it is more compatible withe the proposed hybrid similarity.

Our HyNet architecture is based on L2-Net [46], which consists of seven convolutional layers and outputs 128-dimensional descriptors. As shown in Fig 3, all Batch Normalisation (BN) [17] layers, except the last one before the final $L_2$ normalisation in the original L2-Net, are replaced with FRN layers. Moreover, as recommended in [44], each FRN is followed by the Thresholded Linear Unit (TLU) instead of the conventional ReLU. Thus, HyNet has the same number of convolutional weights as HardNet [26] and SOSNet [47].

## 4 Experiment

Our novel architecture and training is implemented in PyTorch [32]. The network is trained for 200 epochs with a batch size of 1024 and Adam optimizer [20]. Training starts from

scratch, and the threshold $\tau$ in TLU for each layer is initialised with $-1$. We set $\alpha = 2$ and $\gamma = 0.1$. In the following experiments, HyNet is compared with recent deep local descriptors [3, 46, 26, 47] as well as end-to-end methods [10, 11, 34] on three standard benchmarks [7, 2, 40].

## 4.1 UBC verification

UBC dataset [7] consists of three subset-scenes, namely *Liberty*, *Notredame* and *Yosemite*. The benchmark is focused on the patch pair verification task, *i.e.*, whether the match is positive or negative. Following the evaluation protocol [7], models are trained on one subset and tested on the other two. In Table 1, we report the standard measure of false positive rate at 95% recall (FPR@95) [7] on six train and test splits. We can observe that, while the performance is nearly saturated, HyNet still shows remarkable improvements over previous methods.

| Train | ND | YOS | LIB | YOS | LIB | ND | Mean |
|---|---|---|---|---|---|---|---|
| Test | LIB | | ND | | YOS | | |
| SIFT [23] | 29.84 | | 22.53 | | 27.29 | | 26.55 |
| TFeat [3] | 7.39 | 10.13 | 3.06 | 3.80 | 8.06 | 7.24 | 6.64 |
| L2-Net [46] | 2.36 | 4.70 | 0.72 | 1.29 | 2.57 | 1.71 | 2.23 |
| HardNet [26] | 1.49 | 2.51 | 0.53 | 0.78 | 1.96 | 1.84 | 1.51 |
| DOAP [15] | 1.54 | 2.62 | 0.43 | 0.87 | 2.00 | 1.21 | 1.45 |
| SOSNet [47] | 1.08 | 2.12 | 0.35 | 0.67 | 1.03 | **0.95** | 1.03 |
| **HyNet** | **0.89** | **1.37** | **0.34** | **0.61** | **0.88** | 0.96 | **0.84** |

Table 1: Patch verification performance on the UBC phototour dataset. Numbers denote false positive rates at 95% recall(FPR@95). ND: Notredame, LIB: Liberty, YOS: Yosemite.

## 4.2 HPatches matching

HPatches dataset [2] evaluates three tasks, patch verification, patch retrieval, and image matching for viewpoint and illumination changes between local patches. Based on different levels of geometric noise, the results are divided into 3 groups: *easy*, *hard*, and *tough*.

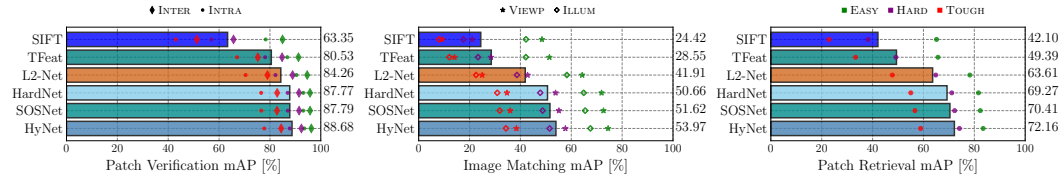

Figure 4: Results on test set 'a' of HPatches [2]. HyNet outperforms the state-of-the-art SOSNet [47] and other local image descriptors in all metrics on this benchmark.

We show the results in Fig. 4, where all models are trained on *Liberty*, which is the protocol proposed in [2]. HyNet improves the MAP from the previous state-of-the-art SOSNet [47] by a large margin, *i.e.*, **0.89**, **2.35**, and **1.75** for the three tasks. Note that the improvement of SOSNet over its predecessor HardNet [26] was 0.03, 0.96, and 1.14 at the time of its publication.

## 4.3 ETH Structure from Motion

ETH SfM benchmark [40] evaluates local descriptors in the task of Structure from Motion (SfM) for outdoor scenes. To quantify the SfM quality, in Table 2, we follow the protocol from [40] and report the number of registered images, reconstructed sparse and dense points, mean track length, and mean reprojection error. First, we compare HyNet with HardNet [26] and SOSNet [47] by using the same local patches extracted from DoG detector, which is presented above the dashed lines. Since the detector is fixed, the results reflect the performance of the descriptors. To ensure a fair comparison, HardNet, SOSNet, and HyNet are all trained on *Liberty* from UBC dataset [7]. In this benchmark, HyNet exhibits significant superiority by registering more images for large scenes and reconstructing more spare points, while the results for the other metrics are on par with top performing descriptors. Next, we compare HyNet to the recent end-to-end methods, namely SuperPoint [10], D2-Net [11] and R2D2 [34].

| | | #Reg. Images | #Sparse Points | #Dense Points | Track Length | Reproj. Error |
|---|---|---|---|---|---|---|
| **Herzjesu 8 images** | SIFT (11.3K) | 8 | 7.5K | 241K | 4.22 | 0.43px |
| | DoG+HardNet | 8 | 8.7K | 239K | 4.30 | 0.50px |
| | DoG+SOSNet | 8 | 8.7K | 239K | 4.31 | 0.50px |
| | DoG+HyNet | 8 | 8.9K | 246K | 4.32 | 0.52px |
| | SuperPoint (6.1K) | 8 | 5K | 244K | 4.47 | 0.79px |
| | D2-Net (13.1K) | 8 | 13K | 221K | 2.87 | 1.37px |
| | R2D2 (12.1K) | 8 | 10K | 244K | 4.48 | 1.04px |
| | Key.Net+HyNet (11.9K) | 8 | 9.4K | 246K | 5.24 | 0.69px |
| **Fountain 11 images** | SIFT (11.8K) | 11 | 14.7K | 292K | 4.79 | 0.39px |
| | DoG+HardNet | 11 | 16.3K | 303K | 4.91 | 0.47px |
| | DoG+SOSNet | 11 | 16.3K | 306K | 4.92 | 0.46px |
| | DoG+HyNet | 11 | 16.5K | 303K | 4.93 | 0.48px |
| | SuperPoint (5.5K) | 11 | 7K | 304K | 4.93 | 0.81px |
| | D2-Net (12.5K) | 11 | 19K | 301K | 3.03 | 1.40px |
| | R2D2 (12.6K) | 11 | 13.4K | 308K | 5.02 | 1.47px |
| | Key.Net+HyNet (11.9K) | 11 | 12.0K | 307K | 7.81 | 0.69px |
| **South Building 128 images** | SIFT (13.3K) | 128 | 108K | 2.14M | 6.04 | 0.54px |
| | DoG+HardNet | 128 | 159K | 2.12M | 5.18 | 0.62px |
| | DoG+SOSNet | 128 | 160K | 2.12M | 5.17 | 0.63px |
| | DoG+HyNet | 128 | 166K | 2.12M | 5.14 | 0.64px |
| | SuperPoint (10.6K) | 128 | 125k | 2.13M | 7.10 | 0.83px |
| | D2-Net (12.4K) | 128 | 178K | 2.06M | 3.11 | 1.36px |
| | R2D2 (13.2K) | 128 | 136K | 3.31M | 5.60 | 1.43px |
| | Key.Net+HyNet (12.9K) | 128 | 100K | 2.11M | 12.03 | 0.74px |
| **Madrid Metropolis 1344 images** | SIFT (7.4K) | 500 | 116K | 1.82M | 6.32 | 0.60px |
| | DoG+HardNet | 697 | 261K | 1.27M | 4.16 | 0.98px |
| | DoG+SOSNet | 675 | 240K | 1.27M | 4.40 | 0.94px |
| | DoG+HyNet | 697 | 337K | 1.25M | 3.93 | 0.98px |
| | SuperPoint (2.1K) | 702 | 125K | 1.14M | 4.43 | 1.05px |
| | D2-Net (7.74K) | 787 | 229K | 0.96M | 5.50 | 1.27px |
| | R2D2 (12.9K) | 790 | 158K | 1.15M | 7.26 | 1.20px |
| | Key.Net+HyNet (9.3K) | 897 | 386K | 1.62M | 5.87 | 1.05px |
| **Gendar- menmarkt 1463 images** | SIFT (8.5K) | 1035 | 338K | 4.22M | 5.52 | 0.69px |
| | DoG+HardNet | 1018 | 827K | 2.06M | 2.56 | 1.09px |
| | DoG+SOSNet | 1129 | 729K | 3.05M | 3.85 | 0.95px |
| | DoG+HyNet | 1181 | 927K | 2.93M | 3.49 | 1.05px |
| | SuperPoint (2.3K) | 1112 | 236K | 2.49M | 4.74 | 1.10px |
| | D2-Net (8.0K) | 1225 | 541K | 2.60M | 5.21 | 1.30px |
| | R2D2 (13.3K) | 1226 | 529K | 3.80M | 6.38 | 1.21px |
| | Key.Net+HyNet (10.6K) | 1259 | 897K | 3.58M | 5.79 | 1.13px |

Table 2: Evaluation results on ETH dataset [40] for SfM.

DoG+HyNet shows significantly better performance on larger scenes, for example, *Madrid Metropolis* and *Gendarmenmarkt*, where it gives over 50% more of reconstructed sparse points in 3D. Note that in the SfM task, the number of registered images and reconstructed points is crucial for the quality of 3D models. Moreover, results also show that HyNet generalises well to different patches provided by the state-of-the-art detector Key.Net [4], where the average track length is increased for a number of scenes.

## 5    Discussion

In this section, we first investigate how each building block of HyNet contributes to the overall performance.

**Ablation Study** is presented in Table. 3, which shows how the $L_2$ norm regularisation term $R_{L_2}$, similarity measure and feature map normalisation affect the performance. Specifically, we train different models on *Liberty* [7] and report average MAP on Hpatches [2] matching task.

| Target | Choice | Other components | MAP |
|---|---|---|---|
| $R_{L_2}$ | ✗ | FRN, $s_H$ | 53.58 |
| | ✓ | FRN, $s_H$ | **53.97** |
| Similarity measure | $d$ | FRN, ✓$R_{L_2}$ | 52.10 |
| | $s$ | FRN, ✓$R_{L_2}$ | 53.19 |
| | $s_H$ | FRN, ✓$R_{L_2}$ | **53.97** |
| Norm type | BN | $s_H$, ✓$R_{L_2}$ | 52.04 |
| | IN | $s_H$, ✓$R_{L_2}$ | 52.47 |
| | FRN | $s_H$, ✓$R_{L_2}$ | **53.97** |
| Descriptor type | | Hard+FRN | 51.80 |
| | | SOSNet+FRN | 52.12 |

Table 3: Ablation of HyNet's components.

First, we can see that $R_{L_2}$ helps to boost the performance, justifying our intuition that it optimises the network to be robust to intensity changes. Next, we compare $s_H$ against $s$ and $d$ in Eqn. (7), where the best results (through grid search for optimal margin) for each similarity are reported. $s_H$ improves from $s$ and $d$ by **1.87** and **0.78** respectively, indicating its effectiveness in balancing the gradient magnitude obtained from the positive and

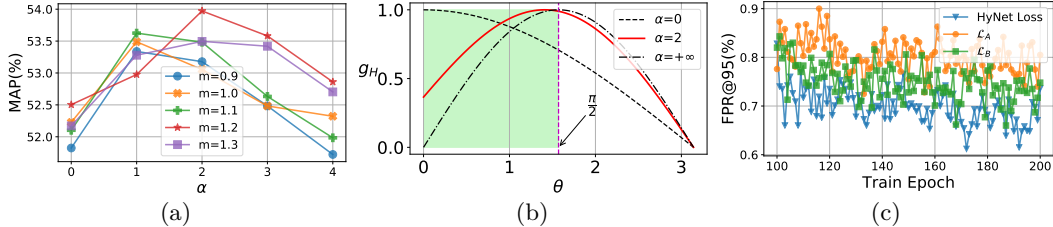

Figure 5: (a) Effect of parameter $\alpha$ in the proposed hybrid loss. (b) Gradient magnitude of the proposed HyNet loss for different $\alpha$. (c) Comparison of the proposed loss to other variants that combine the inner product and $L_2$ loss.

negative samples. Finally, Filter Response Normalisation (FRN) [44] is compared to Batch Normalisation (BN) [17] and Instance Normalisation(IN) [48], where the network with BN is used by previous methods [46, 26, 47, 15]. FRN surpasses BN and IN by at least **1.5**, which demonstrates the advantage of $L_2$ normalising the intermediate feature maps. Above all, by integrating $R_{L_2}$, $s_H$ and FRN together, we achieve the best result. Furthermore, to show that FRN is more compatible with the our proposed hybrid similarity, we retrain HardNet and SOSNet with HyNet architecture. As shown, HyNet gains an MAP improvement of **1.93** from FRN, whereas the numbers for HardNet and SOSNe are 1.33 and 1.10 respectively.

**Effect of $\alpha$ and $m$** is investigated with grid search and reported in Fig. 5(a), where HyNet reaches top performance with $\alpha = 2$ and $m = 1.2$. Furthermore, we plot the gradient magnitude $g_H|s'_H(\theta)|$ in Fig. 5(b) by varying $\alpha$. As seen, the curve of $\alpha = 2$ is in between $\alpha = +\infty$ for $g_s(\theta)$ and $\alpha = 0$ for $g_d(\theta)$, balancing the contributions from positives and negatives.

**Other possible solutions** for using different metrics for the positives and negatives include:

$$\mathcal{L}_A = \frac{1}{N} \sum_{i=1}^{N} \max(0, m_A + s(\theta_i^+) - d(\theta_i^-)),$$

$$\mathcal{L}_B = \frac{\alpha}{N} \sum_{i=1}^{N} \max(0, m_{B_1} + s(\theta_i^+) - s(\theta_i^-)) + \frac{1}{N} \sum_{i=1}^{N} \max(0, m_{B_2} + d(\theta_i^+) - d(\theta_i^-)).$$

(9)

$\mathcal{L}_A$ uses $s$ for positives while $d$ for negatives, which is intuitively the most direct approach for adaptive gradient magnitude. Meanwhile, $\mathcal{L}_B$ stacks two triplet losses, where $m_{B_1}$ and $m_{B_2}$ are the two margins. We conduct grid search for $\mathcal{L}_A$ and $\mathcal{L}_B$, and set $m_A = 1.0$, $\alpha = 2.0$, $m_{B_1} = 0.9$ and $m_{B_2} = 1.2$. Following [47], we compare their training curves with our HyNet loss in Fig. 5(c), where networks are trained on *Liberty* and FPR@95 are average on *Notredame* and *Yosemite*. As shown, our HyNet loss using $s_H$ surpasses the other two solutions. Worth noting, that direct combination in $\mathcal{L}_A$ does not show an advantage. We believe that the triplet loss with a linear margin does not fit well the nonlinear transformation between $s$ and $d$, *i.e.*, $d = \sqrt{2(1-s)}$, but we leave it for future investigation. Meanwhile, stacking triplet losses with different similarity measures is also sub-optimal, which further justifies the effectiveness of the proposed hybrid similarity.

## 6   Conclusion

We have introduced a new deep local descriptor named HyNet, which is inspired by the analysis and optimisation of the descriptor gradients. HyNet further benefits from a regularisation term that constrains the descriptor magnitude before $L_2$ normalisation, a hybrid similarity measure that makes different contributions from positive and negative pairs, and a new network architecture which $L_2$ normalises the intermediate feature maps. Empirically, HyNet outperforms previous methods by a significant margin on various tasks. Moreover, a comprehensive ablation study is conducted revealing the contribution of each proposed component on its final performance.

## Broader Impact

Local feature descriptors and gradient based optimization are crucial components in a wide range of technologies such as stereo vision, AR, 3D reconstructions, SLAM, among others. As such, the proposed approach improves the quality of results within these technologies, which are typically used in various applications including smartphone apps for image processing, driver-less cars, robotics, AR headsets. Its societal impact potential is within these applications, in particular the reliability of the technologies behind, which our approach contributes to. Similarly, any ethical issues are also associated with the applications as our approach cannot be used independently of a larger system.

## Acknowledgments and Disclosure of Funding

This research was supported by UK EPSRC IPALM project EP/S032398/1.

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
