[Supplementary Material]

# HyNet: Local Descriptor with Hybrid Similarity Measure

**Anonymous Author(s) PaperID 1932**

## 1 Image Matching Challenge 2020

We further evaluate HyNet on the newly proposed Image Matching Challenge[1] (IMC) dataset [1]. It consists of two tasks, namely wide-baseline stereo and multi-view reconstruction. Since the ground truth for the test set is not released, we report the performance on the validation set. For fair comparison, we use Key.Net [2] as the detector and compare HyNet with two other state-of-the-art descriptors, HardNet [3] and SOSNet [4]. The evaluation protocol is with a maximum of 2048 keypoints per image and standard descriptor size (512 bytes). We use DEGENSAC [5] for geometric verification, and nearest-neighbour matcher with first-to-second nearest-neighbour ratio test for filtering false-positive matches. Please refer to [1] for exact details of the challenge's settings.

| | mAA (%) | | |
|---|---|---|---|
| | Stereo | Multi-View | Average |
| HardNet [3] | 63.40 | 74.41 | 68.91 |
| SOSNet [4] | 63.41 | 74.51 | 68.96 |
| HyNet | **64.07** | **74.84** | **69.46** |

Table 1: Mean Average Accuracy (mAA) at $10°$ on IMC dataset [1].

As can be seen from Table 1, HyNet surpasses the previous state-of-the-art methods HardNet and SOSNet on both tasks, which further validates its effectiveness.

## 2 Integrating HyNet with SOSR

In this section, we test HyNet by combining it with the Second Order Similarity Regularisation (SOSR) proposed in [4], results are shown in Table 2 and Fig. 1. As shown, HyNet generalises well with the extra supervision signal from SOSR, indicating its potential of being further boosted by other third-party loss terms.

| Train | ND | YOS | LIB | YOS | LIB | ND | Mean |
|---|---|---|---|---|---|---|---|
| Test | LIB | | ND | | YOS | | |
| SIFT [6] | 29.84 | | 22.53 | | 27.29 | | 26.55 |
| HardNet [3] | 1.49 | 2.51 | 0.53 | 0.78 | 1.96 | 1.84 | 1.51 |
| SOSNet [4] | 1.08 | 2.12 | 0.35 | 0.67 | 1.03 | 0.95 | 1.03 |
| HyNet | **0.89** | **1.37** | 0.34 | 0.61 | 0.88 | 0.96 | 0.84 |
| HyNet+SOSR [4] | 0.91 | 1.62 | **0.31** | **0.54** | **0.78** | **0.73** | **0.82** |

Table 2: Patch verification performance on the UBC phototour dataset. Numbers denote false positive rates at 95% recall(FPR@95). ND: Notredame, LIB: Liberty, YOS: Yosemite.

Figure 1: Results on test set 'a' of HPatches [7]. Colour of the marker indicates EASY, HARD, and TOUGH noise. The type of marker corresponds to the variants of the experimental settings.

## Footnotes

[1]`https://vision.uvic.ca/image-matching-challenge/benchmark/`