[Reviews · NeurIPS 2020]

Review 1

Summary and Contributions: This work tackles the area of learning local patch descriptors via deep learning frameworks. It is the area that has been explored a lot recently, with some well established and well performing methods. Authors propose few incremental contributions, compared to those methods: (i) using a hybrid similarity measure instead of inner product or L2-distance-based similarity; (ii) regularization term; (iii) novel network architecture. Finally, the resulting deep local descriptors achieve state-of-the-art performance on all well-established evaluation benchmarks.

Strengths: + I really like the "Gradient Analysis" section, that properly motivates incremental contributions of this work. In fact, I find this part to be the strongest contribution of the paper and should be added as a main one. + Experiments are performed on all well established benchmarks with comparison against proper related work and state-of-the-art approaches + Ablation study gives a really nice overview into the performance of each contribution/part, and beyond + This works clearly sets a new state-of-the-art performance

Weaknesses: - Contributions that are claimed are somewhat weak. In fact, I think that main contribution is in the gradient analysis (as I mentioned in Strengths) and all of the three claimed contributions can be bundled into one "incremental improvements of previous works". Namely: C1: L2-regularization is interesting, but ablation study shows it has the smallest effect on the performance; C2: hybrid similarity measure is a simple combination of two established similarity measures, and it additionally adds another hyper-parameter (alpha) that seems to be very sensitive to setup (Fig5(a) shows that setting lower alpha reduces performance by 1mAP, and setting higher reduces by 0.5 mAP); C3: novel architecture is actually almost identical architecture as [21,22] with addition of FRN block from [36] (ablation study shows this gives the most increase to performance), so it more of a practical combination of previous work than actual contribution. Minor: - Color code Tab 2 with 1st, 2nd and 3rd best result for each column, to be easier for reader to follow, it is a pretty big and unreadable table

Correctness: Yes, claims and experimentation section seems to be in order. I have some remarks about contribution claims that I detail in Strengths and Weaknesses.

Clarity: Paper is well written and easy to read.

Relation to Prior Work: To a careful reader, the connection to prior work is clearly stated.

Reproducibility: Yes

Additional Feedback: I have some problems with paper's contributions, as they seem too incremental and too weak. However, I find the analysis that supports the design choices to be very interesting and results speak for themselves. Because of that, I feel like the paper is interesting for general public and local descriptor learning community and I am leaning towards acceptance of the paper. ** After reading rebuttal and all reviews, I am comfortable with my original rating, and will keep it. **


Review 2

Summary and Contributions: This paper deals with the problem of learning local image descriptors using deep networks. The paper advocates to use 1) L2 normalization for the final descriptors; 2) a hybrid similarity by a weighted combination of the L2 distance and the cosine similarity; 3) filter response normalization (FRN) after each layer of the CNNs instead of batch normalization or instance normalization. A triplet loss function is adopted in the end for learning the descriptors. While the hybrid similarity is partially motivated by a careful analysis of the gradient of both similarities/distances, the other setting are more empirical or standard. Empirical experiments on both patch datasets (UBC and HPatch dataset) and 3D reconstruction datasets (ETH dataset).

Strengths: + The paper is clearly presented and the motivation on using a hybrid similarity to facilitate balanced learning between hard and easy learning samples are well conducted. + The experimental results looks to be solid in three datasets. + The paper is a solid execution combining largely known peieces together for the task of learning local image descriptors.

Weaknesses: - The only weakness is that there are not much exciting new knowledge, in terms of learning, revealed.

Correctness: The claims made are correct, and empirical methodology followed conventions on the adopted datasets.

Clarity: The paper is easy to read and well written.

Relation to Prior Work: The relation to prior work is clearly discussed.

Reproducibility: Yes

Additional Feedback: The paper in overall is a solid execution. But I am afraid the new knowledge advanced here is rather limited. After reading the rebuttal, my rating remains the same. The paper is overall a good execution, it is just the overall knowledge advancement is limited.


Review 3

Summary and Contributions: This paper proposes HyNet for learning local feature descriptors, novelties are 1) hybrid similarity measure, 2) L2 regularization, 3) architecture where all intermediate feature maps are L2-normalized. These choices are justified by an analysis of L2 normalization in gradient-based learning. Results are presented for patch matching, image matching, and 3D reconstruction.

Strengths: + Detailed analysis of the effects of L2 normalization in gradient-based feature learning. The proposed loss is a natural follow-up to the analysis. + The empirical results are impressive, I believe these are the current SOTA results in local feature descriptors.

Weaknesses: - Although the paper strongly advocates the novel loss design, the network architecture and FRN layers appear to play an important role. Which contributes more? I don't think the ablation study in Section 5 is quite sufficient to paint the whole picture. From Table 3, presumably (BN + s_H + R_L2) corresponds to using L2Net architecture with the new loss, which achieves 52.04 MAP - marginally better than 51.62 from SOSNet. I think a key entry missing from all the SOTA comparisons is using baseline L2Net architecture (or one that's as close as possible) + proposed loss. - Conversely, another interesting question is, how will the baselines (esp. HardNet and SOSNet) perform if equipped with FRN normalization? - L161: "... based on the analysis in Sec. 2.2 that, similarly to the output descriptors, L2 normalisation needs to be applied to the intermediate feature maps". I checked Sec 2.2 and didn't find an explicit motivation for L2-normalizing all intermediate feature maps. Also, since FRN is employed anyway, is L2 normalization to the feature maps still really necessary?

Correctness: The math in the analysis of L2 normalization is correct - my complaint is that calling both cosine similarity and L2 distance "similarity measures", and using the same kind of shorthand (s) for them, makes things confusing and should probably be avoided. As I commented in Weaknesses, the empirical methodology of presenting the final product that combines a novel loss and a novel architecture makes things less interpretable. It may not be "incorrect", but it renders the core contributions unclear.

Clarity: Regarding writing (and writing only), yes.

Relation to Prior Work: Yes

Reproducibility: Yes

Additional Feedback: Author's response alleviated some of my concerns, but not all. First, confirming my suspicion, the additional evidence suggests FRN can improve all baselines (although the proposed loss achieves the most additional improvement on top). Thus it seems a fair comparison should be against SOSNet+FRN, HardNet+FRN, etc. in all the tables. Reporting the "final product" is fine for eg. a competition, but I was expecting more for a NeurIPS publication. Second, regarding the motivation of L2-normalizing _all_ intermediate feature maps, authors simply referred back to Sec. 2.2, which is unsatisfactory. From the gradient analysis in the paper, I can clearly see an argument for L2-normalizing the _descriptors_, but generalizing to all feature maps felt like a "leap of faith". It's a tough decision as there's much to like about this paper, but my final score would be 5 (down from 6).

[Author Response · NeurIPS 2020]

**Paper 1932.** We thank the reviewers for their work and feedback. We first address the general comments A-D related to the main contributions from **R1**, **R2**, **R4** and then the specific ones.

**A. R1, R2.** Many works have empirically shown that L2 normalisation improves the matching performance of local features, however, our work is the first to reveal and give insight into why L2 normalisation and its gradients are important in the descriptor training and testing. Furthermore, based on these insights, we take advantage of the underlying mechanism of L2 normalization and further improve the matching performance by proposing a novel loss function. Therefore, we agree with **R1** that investigating this problem from the perspective of the gradients is our strongest and novel contribution.

**B. R1.C3, R2, R4.** Following our gradient analysis, we design a new architecture that applies L2 normalisation in the intermediate feature maps. Thus, our work is the first local descriptor that proposes and displays the benefit of using L2 normalisation within the model and not just on the output as in [21,22]. The new components, *i.e.*, the loss function and the network architecture with FRNs, result from the presented analysis and establish a new state-of-the-art in several benchmarks when used together. Also, see our response to **R4**.

**C. R1, R2.** Previous methods have explored different directions such as data sampling (mining), optimisation of descriptor space distribution, or heuristic designs of loss functions. In contrast, we tackle it from a novel perspective by making better use of the gradients, which is validated by the superior results.

**D. R1, R2.** We believe that our work will have a wide impact in other areas, where feature embedding and matching also rely on L2 normalisation but so far lacked theoretical support, *e.g*, in person re-identification, or face recognition.

**R1.C1 Effect of L2-regularisation term ($R_{L2}$).** The test set of HPatches contains 2M samples (1M positives and 1 M negatives), thus given this large number of samples a 0.39 MAP increase is not marginal (See response to **R4**). Moreover, we further investigate the matching MAP increase brought by $R_{L2}$ in HPatches and find an increase of 0.57 and 0.21 for illumination and viewpoint, respectively. It indicates that our regularisation on the descriptor norm makes the network more robust to illumination variations in the input.

**R1.C2 Hyper-parameter $\alpha$.** Despite the sensitivity, comparing Fig. 5(a) and Fig. 4 (matching MAP in the middle), we can see that all choices of $\alpha$ perform either on par or better than the previous SOTA methods (HardNet, SOSNet). Moreover, such peakedness in the MAP curve by varying $\alpha$ indicates that there is an optimal value of $\alpha$ that can well capture the distribution of the training set.

**R4. Architecture and loss.** The improvements brought by individual components are discussed in lines 235-241. The gain from the new similarity ($s_H$) is up to +1.87 MAP. The gain from the new architecture (FRN+TLU) is +1.5 MAP. Note that both new components benefit each other and lead to higher gains when used together.

New results will be added to Table 3 to further expose the improvements, namely HardNet+FRN: 51.89 (+1.33 from the original HardNet+BN) and SOSNet+FRN: 52.12 (+1.10, from the original SOSNet+BN). Note that the increase of HyNet from BN to FRN is +1.93, indicating that our new loss is more compatible with the FRN layer and that both together take better advantage of the gradients.

The baseline L2Net+proposed loss is indeed ($BN + s_H + R_{L2}$) in Table 3. Under the evaluation protocol of the HPatches dataset with a large number of test samples (1M positives and 1M negatives), 1.0 MAP is significant. The improvement of the previous SOTA SOSNet over HardNet was 0.96 MAP. Finally, we acknowledge that the second order regularisation term (SOSR) from SOSNet is beneficial. Table 1 and Fig. 1 in supplementary material show that SOSR further improves the matching MAP of HyNet by 0.15.

**R4. L161 (L2 normalisation within the architecture).** The analysis in Sec. 2.2 shows the benefits of L2 normalisation. We implement it per feature map by the FRN layer, hence, no further L2 normalisation is needed inside the architecture. We will clarify in Sec. 2.2 where and how the normalisation occurs in the network.

**R4. Similarity measure.** We mentioned on page 3 footnote that L2 is a distance metric (or inverse similarity) but we agree about possible confusion and will change the notation.

**R1. Color coding.** We will highlight the top results with color.

[Meta-Review · NeurIPS 2020]

I have read the reviews, the rebuttal, and much of the paper itself. The paper provides an accessible analysis of the role of L2 normalization on two common "similarity measures". The main issue I find with the analysis is that the claim in lines 89-91 is circular reasoning -- is not the role of the analysis to show why L2 normalization is beneficial? The three architectural modifications suggested are reasonably tied with the analysis, and while each, by itself is not groundbreaking, the final method is powerful. Like some of the reviewers, I am concerned that the major contributor seems to be the FRN method. However, I find the rebuttal convincing enough regarding this point. Based on this, I suggest accepting the paper despite the borderline ratings.